# The Impact of Increased CO_2_ and Drought Stress on the Secondary Metabolites of Cauliflower (*Brassica oleracea* var. *botrytis*) and Cabbage (*Brassica oleracea* var. *capitata*)

**DOI:** 10.3390/plants12173098

**Published:** 2023-08-29

**Authors:** Andreea Lupitu, Cristian Moisa, Flavia Bortes, Denisa Peteleu, Mihaela Dochia, Dorina Chambre, Virgiliu Ciutină, Dana Maria Copolovici, Lucian Copolovici

**Affiliations:** Institute for Research, Development and Innovation in Technical and Natural Sciences, Faculty of Food Engineering, Tourism and Environmental Protection, Aurel Vlaicu University of Arad, Elena Drăgoi Street., No. 2, 310330 Arad, Romania; pag.andreea@yahoo.com (A.L.); moisa.cristian@yahoo.com (C.M.); flaviabortes@yahoo.com (F.B.); denisaioanapeteleu@gmail.com (D.P.); dochiamihaela@yahoo.com (M.D.); dorinachambree@yahoo.com (D.C.); virgilciutina@yahoo.com (V.C.); dana.copolovici@uav.ro (D.M.C.)

**Keywords:** climate change, photosynthetic parameters, secondary metabolites, *Brassicaceae*, volatile organic compounds, elevated carbon dioxide

## Abstract

Elevated carbon dioxide and drought are significant stressors in light of climate change. This study explores the interplay between elevated atmospheric CO_2_, drought stress, and plant physiological responses. Two *Brassica oleracea* varieties (cauliflowers and cabbage) were utilized as model plants. Our findings indicate that elevated CO_2_ accelerates assimilation rate decline during drought. The integrity of photosynthetic components influenced electron transport, potentially due to drought-induced nitrate reductase activation changes. While CO_2_ positively influenced photosynthesis and water-use efficiency during drought, recovery saw decreased stomatal conductance in high-CO_2_-grown plants. Drought-induced monoterpene emissions varied, influenced by CO_2_ concentration and species-specific responses. Drought generally increased polyphenols, with an opposing effect under elevated CO_2_. Flavonoid concentrations fluctuated with drought and CO_2_ levels, while chlorophyll responses were complex, with high CO_2_ amplifying drought’s effects on chlorophyll content. These findings contribute to a nuanced understanding of CO_2_–drought interactions and their intricate effects on plant physiology.

## 1. Introduction

In the summer of 2022, Europe experienced the most extraordinary drought in 500 years. Record-breaking heat and limited precipitation led to rivers drying up, wildfires raging, and crop failures. The American West saw its most severe drought in 1200 years, while portions of China’s biggest river, the Yangtze, reached their lowest level since at least 1865 due to harsh temperatures and a severe lack of rainfall [1,2]. More particularly, carbon dioxide levels in the atmosphere (420.99 ppmv) are currently the highest ever recorded by the Hawaii National Oceanic and Atmospheric Administration (NOAA) mountaintop observatory, a rise of 1.8 ppmv since 2021. Since human activities release more carbon dioxide into the atmosphere each year than natural processes can remove, the concentration of carbon dioxide in the atmosphere will continue to rise. In the year 2100, projections from the Intergovernmental Panel on Climate Change (IPPC) models suggest that carbon dioxide concentrations could fall within the range of 800 to 1200 ppmv [3]. Such behavior will result in a global temperature increase of more than 2 °C, extended periods of drought, and intensified occurrences of extreme climate events. Under such conditions, plants adopt different strategies to cope with environmental stresses, such as gene regulation [4], carbon metabolism change [5], and regulation of stomata opening [6,7,8]. The increased rate of photosynthetic carbon fixation by plant leaves is one of the most consistent impacts of higher atmospheric CO_2_ on plants. A previous study showed that the assimilation rate increased for all 13 species and varieties from the *Brassicaceae* family grown at elevated carbon dioxide levels [9]. The concentrations of carbon dioxide play an essential role in the process of controlling the openness of stomata. Plants regulate stomatal opening as a compromise between high photosynthesis and low water loss rates. Plants can keep their photosynthetic rates high while maintaining a relatively low level of stomatal conductance even when CO_2_ concentrations rise [10,11]. Previous studies showed that flavonoid content increases under elevated carbon dioxide conditions, while antioxidant activity, including total phenolic content, decreases [9,12].

On a global scale, climate change often causes widespread drought stress across huge areas. While all plants are affected by drought, each species adopts specific mechanisms that enhance its drought tolerance [13]. *Brassica napus* plants are negatively affected by drought stress in terms of germination, seedling establishment, photosynthetic efficiency, mineral absorption, shoot elongation, seed development, yield, and quality [14]. The simultaneous presence of many abiotic challenges leads to complicated plant responses that cannot be predicted solely on individual stress responses [15]. *Arabidopsis thaliana*, *Triticum aestivum*, *Brassica napus*, and C_3_ grass have all been used in experiments to examine the crosstalk between high levels of carbon dioxide and drought [16,17,18]. Numerous studies have demonstrated that increased atmospheric CO_2_ can reduce the harmful consequences of drought, with the extent of this impact depending on the specifics of the situation [19,20]. A previous study found that increased CO_2_ concentrations do not reduce total evapotranspiration under drought in *Brassica napus* plants compared to current (400 ppmv) concentrations due to increased leaf area that does not buffer transpiration, and the physiological traits decreased similarly under drought stress for all carbon dioxide concentrations [21]. In the case of different tomato genotypes, drought stress inhibited the net photosynthetic rates and decreased stomata opening regardless of CO_2_ concentration and genotype [22]. More particularly, elevated carbon dioxide growth conditions do not affect the recovery time from stress conditions [22]. While high carbon dioxide levels enhanced photosynthesis and water use efficiency in drought-stressed soybean cultivars during the vegetative stage, this effect did not translate to increased yield. In contrast, other cultivars with high-yielding features exhibited higher biomass and grain yield [23,24,25]. Plants respond to biotic and abiotic stresses by producing volatile organic compounds (VOCs) in several chemical classes, including terpenoids, benzenoids/phenylpropanoids, and fatty acid derivatives [26,27]. It has been shown that volatile organic compound emissions increased for plants grown at elevated carbon dioxide [9]. In most cases, a persistent moderate or intense fast drought stress would ultimately result in a significant drop in VOC emissions, with increased emissions after re-watering the affected area [28]. The combination of drought and elevated carbon dioxide has determined a decrease in photosynthetic rate and increased isoprene emission for *Populus deltoides* [29]. Long-term growth at increased CO_2_ concentrations, compared to growth at ambient CO_2_ concentrations, did not reduce the effects of drought on the *P. deltoides* clone [30]. Nocturnal water fluxes were considerably lower in *Eucalyptus saligna* trees growing in dry soil compared to moist soil, indicating that higher CO_2_ in the dry treatment did not alleviate dryness [31]. Studies have shown that higher levels of carbon dioxide improve the water-use efficiency of *C. vulgaris* and increase leaf photosynthesis throughout the growing season but do not appear to provide adequate protection against drought in terms of phenolics accumulation, which are known to defend plants against drought-induced oxidative stress [32]. When CO_2_ levels were increased, the detrimental effect of dryness on total VOC emissions and accumulation of phenolics were mitigated [33].

Cauliflower (*Brassica oleracea* var. *botrytis*) and Green Cabbage (*Brassica oleracea* var. *capitata*) are two important *Brassicaceae* family members cultivated worldwide. Due to the high water requirements of cauliflower and cabbage leaves, drought stress is one of the significant variables impacting plant development and productivity [34].

We have hypothesized that elevated carbon dioxide could influence the capacity of plants from the *Brassicaceae* family to cope with oxidative stress. Our study aims to evaluate the effect of high carbon dioxide concentrations and drought on two plants from the *Brassicaceae* family on the photosynthesis parameters, pigments, and secondary metabolites.

## 2. Results

### 2.1. Effects of Drought Stress on Photosynthetic Characteristics

The effects of drought stress on the photosynthetic parameters of cauliflower (*Brassica oleracea* var. *botrytis*) and cabbage (*Brassica oleracea* var. *capitata*) were assessed by measuring the net assimilation rate (A) and stomatal conductance to water vapor (g_s_) (Figure 1).

On day 1, significant differences were noticed (one-way ANOVA followed by Tukey’s multiple comparisons post hoc test, *p* < 0.05) between the plants grown at 400, 800, and 1200 µmol mol^−1^, as follows: in the case of cauliflower grown at 400 µmol mol^−1^, the net assimilation rate was 10.38 ± 0.42 µmol m^−2^ s^−1^, for cauliflower grown at 800 µmol mol^−1^, the net assimilation rate was 21.64 ± 0.70 µmol m^−2^ s^−1^, and in the case of cauliflower grown at 1200 µmol mol^−1^, the net assimilation rate was 26.47 ± 0.36 µmol m^−2^ s^−1^.

When comparing the cauliflowers grown at the ambient CO_2_ concentration of 400 ppmv with cauliflowers grown at high concentrations of CO_2_ of 800 ppmv and 1200 ppmv, the net assimilation rate increases significantly (*p* < 0.05).

When subjecting cauliflower and cabbage to drought stress, a significant reduction in the net assimilation rate became evident. When comparing cauliflower grown at 400 ppmv, typically hydrated, with cauliflower plants kept under drought stress, as can be seen in Figure 1, a decrease in net assimilation rate was observed from day 2 onward. This constant decrease, from 10.38 ± 0.42 to 0.21 ± 0.02 µmol m^−2^ s^−1^ ca, persisted until day 7 when the plants were watered. Following the re-watering of the plants, there was an upward trajectory in the net assimilation rate, increasing from 0.21 ± 0.02 to 8.35 ± 0.33 µmol m^−2^ s^−1^ by day 10. When evaluating the cauliflower plants grown at 400, 800, and 1200 ppmv that were kept in drought conditions, the net assimilation rate decreased for all plants until day 7 when the plants were re-watered. By day 10, this parameter had increased from nearly 0 to 13.15 ± 0.36 µmol m^−2^ s^−1^ for plants grown at 800 ppmv, and to 15.47 ± 0.26 µmol m^−2^ s^−1^ for plants grown at 1200 ppmv.

In the case of cabbage plants grown at different concentrations of CO_2_, a considerable increase in net assimilation rate was observed. For cabbage plants grown at 400 ppmv, the net assimilation rate was 13.58 ± 0.50 µmol m^−2^ s^−1^; for plants cultivated at 800 ppmv, the net assimilation rate was 19.77 ± 0.69 µmol m^−2^ s^−1^, and for plants grown at 1200 ppmv, the net assimilation rate was 23.41 ± 0.32 µmol m^−2^ s^−1^.

In the case of cabbage, on the first day, differences were also observed between the plants grown at 400, 800, and 1200 ppmv, as follows: in the situation of cabbage grown at 400 ppmv, the net assimilation rate was 13.58 ± 0.50 µmol m^−2^ s^−1^, for cabbage grown at 800 ppmv, the net assimilation rate was 19.77 ± 0.69 µmol m^−2^ s^−1^, and in the case of cauliflower grown at 1200 ppmv, the net assimilation rate was 23.41 ± 0.32 µmol m^−2^ s^−1^. When comparing the cabbage plants grown at 400 ppmv of CO_2_ with cauliflowers grown at high concentrations of CO_2_, 800 ppmv, and 1200 ppmv, the net assimilation rate rises considerably.

When comparing the cabbage grown at 400 ppmv, typically hydrated, with cabbage plants kept under drought stress, as can be seen in Figure 1, a decrease in net assimilation rate can be observed from day 2 onward (one-way ANOVA followed by Tukey’s multiple comparisons post hoc test, *p* < 0.05). This constant decrease from 13.58 ± 0.50 to 1.65 ± 0.14 µmol m^−2^ s^−1^ can be observed until day 7 when the plants were re-watered. After the re-watering of the plants, the net assimilation rate increased from 1.65 ± 0.14 to 9.75 ± 0.26 µmol m^−2^ s^−1^ until day 10.

When evaluating the cabbage plants grown at 400, 800, and 1200 ppmv that were kept in drought conditions, the net assimilation rate decreased for all plants until day 7 when the plants were re-watered. Afterward, by day 10 this parameter increased from nearly 1.18 ± 0.10 to 16.76 ± 0.35 µmol m^−2^ s^−1^ for plants grown at 800 ppmv, and 17.51 ± 0.26 µmol m^−2^ s^−1^ for plants grown at 1200 ppmv.

The drought stress applied to cauliflower and cabbage plants also affects the stomatal conductance to water vapor (gs). As can be observed in Figure 2, the stomatal conductance to water vapor has different values depending on the treatment used.

In the case of cauliflower plants grown at different CO_2_ concentrations of 400, 800, and 1200 ppmv, the stomatal conductance to water vapor exhibited different values. In the case of cauliflower grown at 400 ppmv, the stomatal conductance to water vapor was 67.95 ± 1.39 mmol m^−2^ s^−1^. For cauliflower grown at 800 ppmv, the stomatal conductance to water vapor was 79.20 ± 0.76 mmol m^−2^ s^−1^, and in the case of cauliflower grown at 1200 ppmv, the stomatal conductance to water vapor was 79.61 ± 0.27 mmol m^−2^ s^−1^.

For cauliflower plants grown at 400 ppmv that were subjected to drought stress, the stomatal conductance to water vapor decreased from 67.95 ± 1.39 mmol m^−2^ s^−1^ to 1.11 ± 0.10 mmol m^−2^ s^−1^ at day 7. After the plants were re-watered, the stomatal conductance to water vapor increased to 54.28 ± 0.82 mmol m^−2^ s^−1^. This parameter recovered but did not reach the value observed on day 1. The same pattern was observed for plants grown at 800 ppmv and for those grown at 1200 ppmv. For plants grown at 800 ppmv, the stomatal conductance to water vapor also decreased starting from day 7, going from 79.20 ± 0.76 mmol m^−2^ s^−1^ to 1.13 ± 0.07 mmol m^−2^ s^−1^. After the re-watering treatment, it increased to 45.48 ± 0.45 mmol m^−2^ s^−1^. In the case of plants grown at 1200 ppmv, the value of stomatal conductance to water vapor dropped from 79.71 ± 0.27 mmol m^−2^ s^−1^ to 1.33 ± 0.21 mmol m^−2^ s^−1^ by day 7. Following the re-watering, it rose to 46.44 ± 0.40 mmol m^−2^ s^−1^.

The stomatal conductance to water vapor values from cabbage plants grown at different CO_2_ concentrations of 400, 800, and 1200 ppmv showed no statistically significant differences (*p* = 0.286).

In the case of drought treatment applied to cabbage plants grown at 400 ppmv, the stomatal conductance to water vapor showed a decrease until day 7, reaching 2.10 ± 0.38 mmol m^−2^ s^−1^, followed by an increase to 73.01 ± 1.37 mmol m^−2^ s^−1^ after re-watering. For cabbage plants grown at 800 ppmv, this parameter also decreased until day 7, to 1.69 ± 0.34 mmol m^−2^ s^−1^, and after the re-watering treatment, an increase to 59.73 ± 0.63 mmol m^−2^ s^−1^. In the case of cabbage plants grown at 1200 ppmv, the drought treatment triggered a decrease in stomatal conductance to water vapor until day 7, at 1.78 ± 0.01 mmol m^−2^ s^−1^. After the re-watering, there was an increase to 60.50 ± 0.20 mmol m^−2^ s^−1^.

### 2.2. The Emission of Volatile Organic Compounds from Plants under Drought Stress

The drought stress subjected to cauliflower and cabbage plants also affects the volatile organic compound (VOC) emissions. Figure 3 shows that VOC emissions vary according to the severity of drought treatment.

On day 1, all plants had very low VOC emissions despite their growth conditions (*p* = 0.985). Cauliflower plants grown at 400 ppmv emitted 0.04 ± 0.01 nmol m^−2^ s^−1^ of VOCs, cauliflower plants grown at 800 ppmv emitted 0.06 ± 0.01 nmol m^−2^ s^−1^, and cauliflower plants grown at 1200 ppmv emitted 0.06 ± 0.01 nmol m^−2^ s^−1^.

If the drought treatment is applied to cauliflowers grown at 400 ppmv, VOC emissions rise until day 7, at 1.05 ± 0.21 nmol m^−2^ s^−1^; for cauliflowers grown at 800 ppmv, the emissions increase to 1.64 ± 0.11 nmol m^−2^ s^−1^; and for cauliflower plants grown at 1200 ppmv VOC emissions increase to 1.67 ± 0.18 nmol m^−2^ s^−1^.

After the re-watering of cauliflowers, VOC emissions decreased for all plants. For cauliflowers grown at 400 ppmv, the emissions decreased to 0.71 ± 0.18 nmol m^−2^ s^−1^; for plants grown at 800 ppmv, the emissions decreased to 1.04 ± 0.14 nmol m^−2^ s^−1^; and for plants grown at 1200 ppmv, the emissions decreased to 1.02 ± 0.11 nmol m^−2^ s^−1^. When grown at 400 ppmv, cabbage plants emitted 0.02 ± 0.01 nmol m^−2^ s^−1^ of volatile organic compounds; when grown at 800 ppmv, VOC emissions were 0.03 ± 0.01 nmol m^−2^ s^−1^; and when grown at 1200 ppmv, VOC emissions were 0.03 ± 0.01 nmol m^−2^ s^−1^.

When the drought treatment was applied to 400 ppmv cabbage plants, the VOC emissions increased until day 7, reaching 0.57 ± 0.11 nmol m^−2^ s^−1^; for 800 ppmv cabbage plants, the emission increased to 0.62 ± 0.11 nmol m^−2^ s^−1^; and for 1200 ppmv cabbage plants, the VOC emissions increased to 0.74 ± 0.12 nmol m^−2^ s^−1^.

After the re-watering stage, VOC emissions decreased to a range of 0.33–0.48 nmol m^−2^ s^−1^.

### 2.3. The Changes in Total Phenolic Content for Plants under Drought Stress

The total phenolic content is different when comparing cauliflower and cabbage plants grown at varying concentrations of CO_2_, as can be seen in Figure 4.

In the case of cauliflower, differences can be observed between plants grown at 400 ppmv and plants grown at elevated carbon dioxide (one-way ANOVA followed by Tukey’s multiple comparisons post hoc test, *p* < 0.05). The total phenolic content varies between 26.8 ± 2.8 mg gallic acid equivalents/g DW and 20.2 ± 4.1 mg gallic acid equivalents/g DW. As shown in Figure 4, as the concentration of CO_2_ increases, the total phenolic content decreases. Thus, the highest quantity is obtained at 400 ppmv. If drought stress is applied, the total phenolic content increases for all concentrations of CO_2_ until day 7 (with some variations). In the case of cauliflower grown at 400 ppmv, the total phenolic content exhibits a significant increase (*p* < 0.05) rising from 26.8 ± 2.8 mg gallic acid equivalents/g DW on day 1 to 30.5 ± 1.3 mg gallic acid equivalents/g DW on day 7. The same pattern was followed by plants grown at elevated carbon dioxide, although the trend is unclear. Following re-watering, the total phenolic content decreases once again for all studied plants, approaching initial levels.

With cabbage, many apparent differences between plants cultivated at 400 ppmv, 800 ppmv, and 1200 ppmv can be seen. The total phenolic content ranges between 15.7 mg of gallic acid equivalents/g DW and 19.1 mg of gallic acid equivalents/g DW. The total phenolic content decreases with rising CO_2_ concentrations, as can be seen in Figure 4, and the maximum amount is found at 400 ppmv. The phenolic content increases incrementally under drought stress for all CO_2_ concentrations until day 7. In plants grown at 400 ppmv, the total phenolic content increased from 19.1 ± 1.9 mg gallic acid equivalents/g DW to 49.9 ± 3.1 mg gallic acid equivalents/g DW; in plants grown at 800 ppmv, the total phenolic content increased from 15.9 ± 0.2 mg gallic acid equivalents/g DW to 28.7 ± 0.8 mg gallic acid equivalents/g DW; and in plants grown at 1200 ppmv, the total phenolic content increased from 15.7 ± 1.1 mg gallic acid equivalents/g DW to 29.4 ± 2.0 mg gallic acid equivalents/g DW. For all the plants that were investigated, the total phenolic content declined after the re-watering although not to the same levels as before the application of drought stress.

### 2.4. The Changes in the Flavonoid Content for Plants under Drought Stress

The flavonoid content is not statistically different (*p* = 0.119) between cauliflower plants grown at varying concentrations of CO_2_, as can be seen in Figure 5. The values range from 5.1 to 5.9 mg rutin equivalents/g DW. In the case of drought stress application, the values for the flavonoid content rise until day 7. For cauliflower plants grown at 400 ppmv, the flavonoid content increases from 5.1 ± 0.3 to 9.6 ± 1.0 mg rutin equivalents/g DW; for plants grown at 800 ppmv, the content increases from 5.4 ± 0.1 to 10.9 ± 1.1 mg rutin equivalents/g DW; and for plants grown at 1200 ppmv, the content from 5.9 ± 0.4 to 9.6 ± 1.0 mg rutin equivalents/g DW (*p* < 0.05). After the re-watering treatment, the flavonoid content decreases, but the levels are higher—nearly double—than on day 1.

Figure 5 illustrates that when comparing cabbage plants grown at various CO_2_ concentrations, there is no significant change in the flavonoid content (*p* = 0.086). The values range from 2.7 to 3.7 mg of rutin equivalents/gDW—the flavonoid content increases till day 7 when drought stress is applied. The flavonoid content in cabbage plants grown at 400 ppmv rises from 3.7± 0.7 to 8.6 ± 1.30 mg rutin equivalents/g DW. The flavonoid content declines after the re-watering treatment, but the levels are still greater than on day 1. There are no significant modifications in flavonoid content for plants grown at 800 and 1200 ppmv CO_2_.

### 2.5. The Changes in Pigment Concentrations of Cauliflower and Cabbage Plants under Drought Stress

Figure 6 shows that the chlorophyll content varies between cauliflower plants grown at different CO_2_ concentrations. The content of chlorophyll *a* ranged from 765 ± 14 to 947 ± 7 mg m^−2^. When drought stress is applied, the chlorophyll *a* content decreases for all CO_2_ concentrations. At 400 ppmv, the chlorophyll *a* content decreased from 766 to 502 mg m^−2^; at 800, the decrease was from 921 to 642 mg m^−2^; and at 1200, the decrease was from 947 to 613 mg m^−2^. After the re-watering treatment, the chlorophyll content decreases for all CO_2_ concentrations.

As can be seen in Figure 6, in the case of cabbage, the chlorophyll *a* content for plants grown at 400 differs from that of those grown at 800 and 1200 ppmv (*p* < 0.05). The chlorophyll *a* content increases from 435 ± 3 to 494 ± 8 mg m^−2^ for plants grown at 1200 ppmv. In the case of drought treatment application, the chlorophyll *a* content decreases until day 7 for all plants. After the re-watering treatment, the chlorophyll *a* content increased for all plants; however, the levels remained smaller than those of the unstressed plants.

When the chlorophyll *b* content is analyzed for cauliflower and cabbage plants, the same pattern is observed as that for chlorophyll *a* (Figure 7). Plants grown under different conditions have significantly different chlorophyll *b* content (one-way ANOVA followed by Tukey’s multiple comparisons post hoc test, *p* < 0.05). The chlorophyll *b* content significantly decreases in the case of drought (*p* < 0.05 for all treatments). At the same time, the effect is more pronounced for *Brassica oleracea* var. *botrytis* plants than for *Brassica oleracea* var. *capitata* plants.

## 3. Discussion

Many studies have suggested that increased atmospheric CO_2_ may reduce the detrimental impacts of drought, with the precise result varying depending on the plant species and the severity of the treatment applied [19,20]. It has been demonstrated that plants grow better in elevated carbon dioxide and have higher assimilation rates [10]. In our case, it can be seen from Appendix A that plants grown at higher carbon dioxide concentrations have more plant material than control plants (grown at 400 ppmv). A prior study found that increased CO_2_ concentrations do not lower total evapotranspiration under drought in rapeseed plants compared to concentrations of CO_2_ of 400 ppmv because an increase in leaf area does not help protect transpiration, and the physiological traits decreased similarly under drought stress for all carbon dioxide concentrations [21]. In our case, the assimilation rates for plants grown at elevated carbon dioxide drop faster than for plants grown at 400 ppmv. Drought can impact the physical integrity of the components of the photosynthetic apparatus, compromising electron transport capacity and reducing the efficiency of photosynthetic activity [35]. Such behavior could be linked with drought impacting the activation state of nitrate reductase under both raised and ambient CO_2_ settings. However, the effect was more significant under ambient CO_2_ than at increased CO_2_ levels. Increased carbon dioxide levels facilitate the redistribution of surplus nitrogen and other crucial substances away from the photosynthetic machinery towards other growth-constraining mechanisms, such as antioxidant defense [36].

Furthermore, increasing carbon dioxide growth parameters does not influence recovery during stressful situations [22]. High carbon dioxide increases photosynthesis and water-use effectiveness during the vegetative stage under drought stress. Additionally, distinct behaviors are observed in cauliflower and green cabbage. Chavan et al. observed no discernible disparity in the stomatal conductance of wheat plants when cultivated and assessed under varying carbon dioxide concentrations (normal and elevated carbon dioxide) [37]. Conversely, Zinta et al. documented an elevation in the stomatal conductance of *Arabidopsis thaliana* plants when exposed to high carbon dioxide levels under non-stressful conditions [38]. The limited availability of water induces the closing of stomata, hence exerting a detrimental impact on the gas exchange process [39]. The variation in plant growth can be attributed to factors such as plant species and developmental stage, which may be associated with limitations in water or nutrient availability [22]. In the recovery period, the plants grown in elevated carbon dioxide have lower stomata conductance than plants grown in normal conditions.

The emission of monoterpenes generally increases with the duration of drought. Some authors have found that cabbage plants grown at increased CO_2_ concentrations have had significantly lower total monoterpene emissions per dry shoot weight than plants grown at normal CO_2_ [40]. Conversely, other authors have found that oak trees have increased monoterpenes emissions [41]. Such differences can be attributed to the specificity of the species. In the case of cabbage and cauliflower plants, the emissions are very low for plants without drought stress, but there are significant differences for plants under drought stress. An increase in emissions could be due to the build-up of intercellular concentration of isoprenoids [42]. At the same time, the specific solubility could explain the higher emission for plants grown under elevated carbon dioxide concentrations. Subsequently, the gas–liquid phase balance of the many volatile chemicals inside the leaf regulates the emission from the storage pools and non-specific storage [43].

In general, drought led to an increase in the total amount of polyphenols and individual polyphenols, and this rise was much more prominent when there was a significant water shortage. Conversely, polyphenol levels and ethylene biosynthesis decrease for plants grown at elevated carbon dioxide [44]. Such antagonistic effect has been found for both species, but the *Brassica oleracea* var. *capitata* plants significantly increased in phenol levels on the last day of drought. The same results have been found for prolonged drought in *Brassica rapa* [45]. During the recovery period, the concentration of polyphenols decreases almost to the same value as those at the beginning of the experiments. This could be explained by the protective role of those compounds and the limitation of carbon available for secondary metabolites production.

The total flavonoid concentration for plants grown at high carbon does not differ significantly from that of plants grown at 400 ppmv. A significant difference has been obtained for plants grown at different carbon dioxide concentrations. It has been shown that elevated carbon dioxide increased flavonoid concentration in *Zingiber officinale* [46] and *Betula pendula* [47]. The concentration of flavonoids increased in green, fully formed flag leaves of *Triticum aestivum* when exposed to elevated carbon dioxide conditions (550 ppmv), with isoorientin levels rising by 46% and tricin levels increasing by 55%. However, these changes in flavonoid concentrations were no longer seen in flag leaves that had reached maturity and were undergoing senescence [48]. The flavonoid concentration increases at the end of the drought period. In our case, after the drought, a higher concentration of flavonoids has been found for plants growing at 400 ppmv CO_2_. After the recovery time, the flavonoid concentration decreases for all treated plants, which could be explained by the fact that decreasing stress power reduces the flavonoid concentration as flavonoids are influential ROS scavengers [49].

Reduced overall chlorophyll concentration and a shift in the chlorophyll a/b ratio are typical responses of plants to drought [50]. Chlorophyll loss due to photo-oxidation of pigments or chlorophyll degradation is a common sign of oxidative stress under drought conditions [51]. Conversely, elevated carbon dioxide increases photosynthetic pigments and photosynthetic efficiency. An antagonistic effect of this kind could be observed for both *Brassicaceae* plants in our study. Regardless, chlorophyll levels of plants grown in elevated carbon dioxide have been more affected by drought. As drought affects the biochemical process of photosynthesis, the combination of the stomatal limitation, non-stomatal limitation, and chlorophyll content could be responsible for decreasing the photosynthetic activities of plants (especially those grown at elevated carbon dioxide) [52]. After re-watering, the chlorophylls recovered partially in all plants. Such behavior has been shown for some safflower genotypes [53]. More particularly, there is a difference between the chlorophyll contents of green cabbage and cauliflower, which could be explained by a difference in the plants’ capacity to assimilate fertilizers and essential nutrients, as chlorophyll synthesis in the plants is directly related to the availability of those compounds [54].

## 4. Materials and Methods

### 4.1. Plant Material

Green Cabbage (*Brassica oleracea* var. *capitata*, Varza de Buzau (Sem-Luca, Timisoara, Romania) and Cauliflower (*Brassica oleracea* var. *botrytis*, Moldovita F1 (Agrosel, Campia-Turzii, Romania)) seeds were sown in 0.8 L plastic containers containing a mixture of commercial garden soil and quartz sand. The plants were fertilized with foliar (Bionat Plus, Panetone SRL, Timisoara, Romania) and radial (Cropcare 11-11-21, YaraMila^TM^ Cropcare, Yara Mila International, Oslo, Norway) fertilizers. The light intensity at the plant level was 800 mol m^−2^ s^−1^ and was supplied by LED lamps (Hoff, Nürnberg, Germany). Temperatures and relative humidity were maintained at 25/22 °C and 65%, respectively. Plants were grown from seeds at 400 ppmv (control), 800 ppmv, and 1200 ppmv (Appendix A). The vegetation was irrigated daily to the capacity of the soil field. The investigations utilized seven-week-old, non-bolting plants with at least three developed leaves. We utilized the entirely expanded leaves at the same developmental stage for all measurements. Randomization ensured that all plants received the same amount of light. The leaves used for photosynthetic and volatile organic compound measurements were utilized for the biochemical analysis.

### 4.2. Photosynthetic Measurements

As previously reported, a portable gas exchange system (GFS-3000, Waltz, Effeltrich, Germany) was used to determine the photosynthetic parameters [55,56]. The steady-state values of net assimilation (A) and stomatal conductance to water vapor (gs) were calculated using the method described in a previous study [56].

### 4.3. Volatile Sampling and GC–MS Analyses

A flow air sample pump Model 210-1003 MTX (SKC Inc., Houston, TX, USA) was used to collect samples of volatile organic compounds (VOC), which were then analyzed using a Shimadzu TD20 automated cartridge desorber paired with a Shimadzu 2010 Plus GC–MS apparatus (Shimadzu Corporation, Kyoto, Japan) using the method described in a previous study [57]. The VOC sampling and calculation were carried out as described in a previous study [55]. The VOC emission has been expressed in nmol m^−2^ s^−1^ following the procedure described in a previous study [56]. The volatiles were identified by comparing the mass spectrum of individual compounds with the spectra of GC purity external standards and NIST Library’s spectra.

### 4.4. Chromatographic Analysis of Photosynthetic Pigments

The pigments (chlorophyll a, chlorophyll b, and beta-carotene) were extracted in acetone according to the technique reported previously [58], and the quantitative analyses were carried out utilizing the UHPLC-DAD equipment (NEXERA 8030, Shimadzu, Kyoto, Japan) following the method described in a previous study [55]. In order to calculate the level of chlorophyll *a*, chlorophyll *b*, and beta-carotene, pure chromatographic standards (Merck, Darmstadt, Germany) were used.

### 4.5. Flavonoid Content Analysis

A spectrophotometric approach was utilized to read the total flavonoid content [57] accurately. At a wavelength of 434 nm, a reaction mixture including aluminum chloride, sodium acetate, and the sample was measured, and the findings were represented as the number of mg rutin equivalents/DW.

### 4.6. Total Phenolic Content—Folin–Ciocalteu Method

The Folin–Ciocalteu technique, as described in previous studies [57,59], was used to estimate the total phenolic content of the sample, and the findings were reported in terms of mg gallic acid equivalents/DW.

### 4.7. Statistical Analysis and Data Handling

One-way ANOVA and Tukey’s multiple comparisons test were performed using GraphPad Prism version 10.0.2 for Windows (GraphPad Software, San Diego, CA, USA, www.graphpad.com, accessed on 22 July 2023). Data sharing different letters are significantly different (*p* < 0.05), while data sharing the same letters are not significantly different (*p* > 0.05).

## 5. Conclusions

Plants cultivated at elevated carbon dioxide had lower absorption rates than those grown at 400 ppmv CO_2_, perhaps because dryness affects nitrate reductase activation more at ambient CO_2_ than at elevated CO_2_. Under drought stress, increasing carbon dioxide during the vegetative stage improves photosynthesis and water usage efficiency but lowers stomatal conductance after recovery. Plants cultivated at greater CO_2_ concentrations may emit fewer monoterpenes, while the drought period boosts monoterpene emissions. Elevated carbon dioxide decreases polyphenols and ethylene production, while drought increases both total and individual polyphenols. High carbon dioxide boosts flavonoid concentration near the end of the dry phase, but all treated plants’ concentrations fall following recovery. Drought decreases chlorophyll content and can change the a/b ratio, whereas high carbon dioxide improves photosynthetic pigments and efficiency. However, dryness affects chlorophyll content and photosynthetic activity more in plants with increased carbon dioxide.

These findings illustrate the opposing and complicated impacts of increased CO_2_ and drought on critical plant physiological systems such as photosynthesis, gas exchange, and secondary metabolite biosynthesis. The intensity of the drought and the type of plant being studied may affect the results.

## Figures and Tables

**Figure 1 plants-12-03098-f001:**
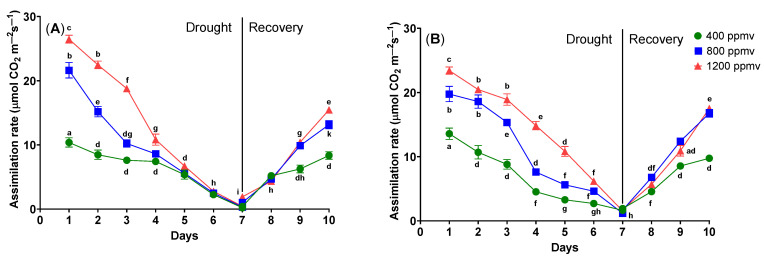
The assimilation rates (A) of *Brassica oleracea* var. *botrytis* (**A**) and *Brassica oleracea* var. *capitata* (**B**) plants grown at three carbon dioxide concentrations in the drought and recovery conditions. Data sharing different letters are significantly different (*p* < 0.05), while data sharing the same letters are not significantly different (*p* > 0.05). The values are averages of three independent measurements.

**Figure 2 plants-12-03098-f002:**
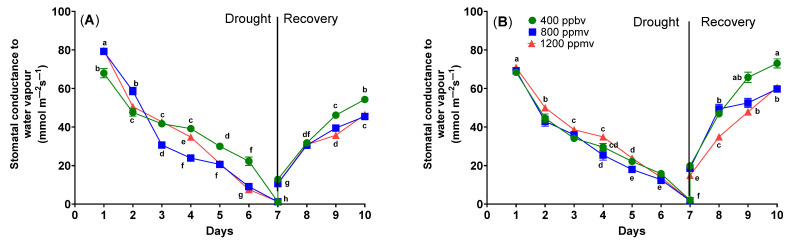
The Stomatal conductance to water vapor (g_s_) of *Brassica oleracea* var. *botrytis* (**A**) and *Brassica oleracea* var. *capitata* (**B**) plants grown at three carbon dioxide concentrations in the drought and recovery conditions. Data sharing different letters are significantly different (*p* < 0.05), while data sharing the same letters are not significantly different (*p* > 0.05). The values are averages of three independent measurements.

**Figure 3 plants-12-03098-f003:**
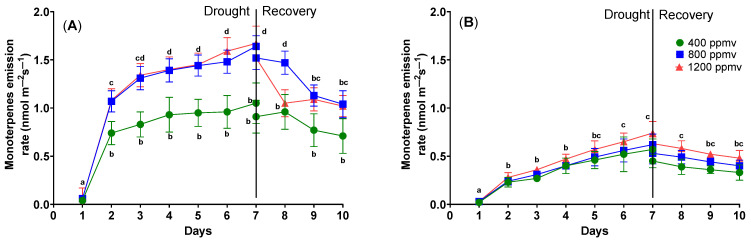
The VOC emission from *Brassica oleracea* var. *botrytis* (**A**) and *Brassica oleracea* var. *capitata* (**B**) plants grown at three carbon dioxide concentrations in the drought and recovery conditions. Data sharing different letters are significantly different (*p* < 0.05), while data sharing the same letters are not significantly different (*p* > 0.05). The values are averages of three independent measurements.

**Figure 4 plants-12-03098-f004:**
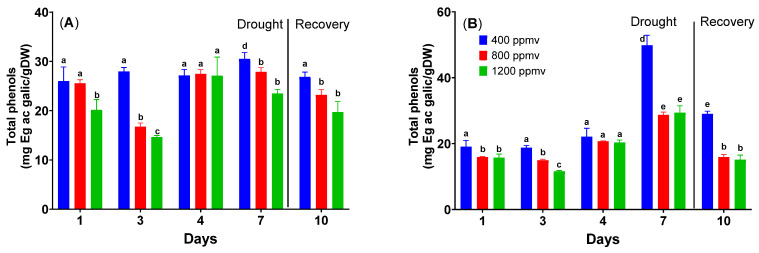
Total phenolic content of *Brassica oleracea* var. *botrytis* (**A**) and *Brassica oleracea* var. *capitata* (**B**) plants grown at three carbon dioxide concentrations in the drought and recovery conditions. Data sharing different letters are significantly different (*p* < 0.05), while data sharing the same letters are not significantly different (*p* > 0.05). The values are averages of three independent measurements.

**Figure 5 plants-12-03098-f005:**
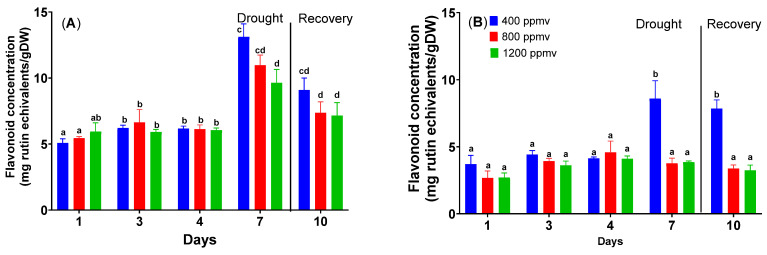
Total flavonoid content of *Brassica oleracea* var. *botrytis* (**A**) and *Brassica oleracea* var. *capitata* (**B**) plants grown at three carbon dioxide concentrations in drought and recovery conditions. Data sharing different letters are significantly different (*p* < 0.05), while data sharing the same letters are not significantly different (*p* > 0.05). The values are averages of three independent measurements.

**Figure 6 plants-12-03098-f006:**
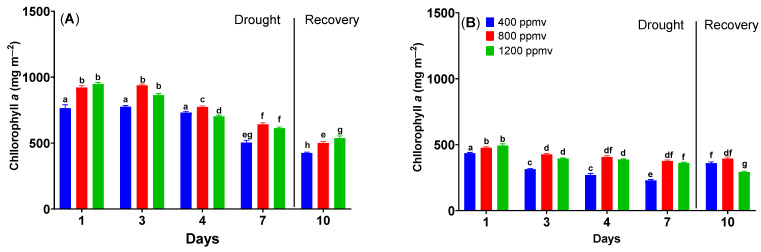
The changes in chlorophyll *a* content of *Brassica oleracea* var. *botrytis* (**A**) and *Brassica oleracea* var. *capitata* (**B**) plants grown at three carbon dioxide concentrations in the drought and recovery conditions. The values are averages of three independent measurements.

**Figure 7 plants-12-03098-f007:**
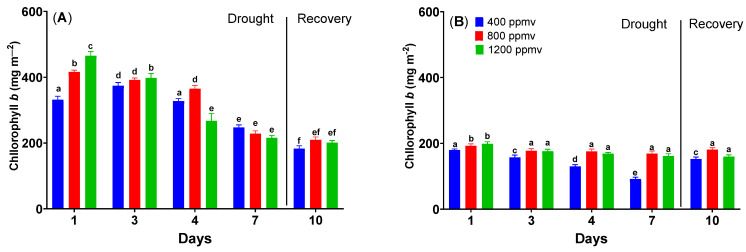
The changes in chlorophyll *b* content of *Brassica oleracea* var. *botrytis* (**A**) and *Brassica oleracea* var. *capitata* (**B**) plants grown at three carbon dioxide concentrations in the drought and recovery conditions. The values are averages of three independent measurements.

## Data Availability

The data presented in this study are available in the article.

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
