# Peer review of "The Impact of Increased CO2 and Drought Stress on the Secondary Metabolites of Cauliflower (Brassica oleracea var. botrytis) and Cabbage (Brassica oleracea var. capitata)"

_plants, 2023, doi:10.3390/plants12173098_

Round 1

Reviewer 1 Report

see letter attached

Author Response

Please see the answer in the attached file.

Reviewer 2 Report

The increase of carbon dioxide and drought are important factors of climate change. Lupitu et al. analysis the carbon dioxide and concentration and the water availability elevation affect photosynthetic parameter, the volatile organic compounds emitted, and photosynthetic pigments from both species of cauliflower and cabbage. The topic is useful, but there are some important problems with the results needs to be resolved.

The phenotypic figures should be supplemented.

Lines 104, 157, 197, 224, 265, 287, 296: The statistical analysis should be used the ANNOVA analysis.

The authors should explain the consistency and inconsistency of the assimilation rate, stomatal conductance, VOC emission, phenolic content, flavonoid content, changes of chlorophyll a, and so on.

Line 158: Stomatal should be stomatal

Line 297: chlorophyll a or chlorophyll b?

In addition, there was no Figure 7 cited in the manuscript.

Moderate editing of English language required

Author Response

(The authors gave the same response as above.)

Round 2

Reviewer 2 Report

The revised manuscript has been improved. But there were still some important problem to be resolved.

There was no Figure S1 cited in the manuscript?

Please explain the relationship of the phenotype and physiological indicators of Green Cabbage (Brassica oleracea var. capitate) and Cauliflower (Brassica oleracea var. botrytis) grown at 400, 800, 1200ppmv in the Figure S1.

There was no control in the Figure S1a, and no 400 ppmv in Figure S1b. Is the Figure S1 true?

Was there three Biological repeated experiments?

There was puzzled in the statistical analysis markers in Figures 1, 2 and 3.

The authors should list the location of each revised point in the revised manuscript.

Minor editing of English language required.

Round 3

Reviewer 2 Report

The Lines described in the coverletter was wrong.

There could be some wrong. Please check it!

"We used 3 different plants grown in the same conditions for every measurement. " It does not mean the biological repeated experiments.

Minor editing of English language required

Author Response

Dear Reviewer,

We have taken all individual Reviewer points seriously into consideration. Our reply to each comment is given below, following the Reviewers' comments and suggestions. We also indicate in the manuscripts which action was taken using Track changes. Anyway, we added a list with all the changes in the MS.

The Lines described in the coverletter was wrong.

A: We used “track changes” to show in the MS modifications. Anyway, please see below the list with the modifications with the lines as in the attached file.

Lines 104, 157, 197, 224, 265, 287, 296: The statistical analysis should be used the ANNOVA analysis.

A: The statistical data have been added L 133, 141, 177, 220, 245, 279, 288, 315, 323

The authors should explain the consistency and inconsistency of the assimilation rate, stomatal conductance, VOC emission, phenolic content, flavonoid content, changes of chlorophyll a, and so on.

A: We add more discussions.

L 386-389, 400-403, 410-420, 446-455, 480-484

Line 158: Stomatal should be stomatal

A: corrected L189

Line 297: chlorophyll a or chlorophyll b?

A: corrected L 370

In addition, there was no Figure 7 cited in the manuscript.

A: Corrected and discussed L376-382

There was no Figure S1 cited in the manuscript?

A: We added the Figure S1’ citation in the MS (L388, L496).

Please explain the relationship of the phenotype and physiological indicators of Green Cabbage (Brassica oleracea var. capitate) and Cauliflower (Brassica oleracea var. botrytis) grown at 400, 800, 1200ppmv in the Figure S1.

A: We added an explanation (L386-389), but the idea of the study has not been to measure phenotype characteristics.

There could be some wrong. Please check it!

A: ?

"We used 3 different plants grown in the same conditions for every measurement. " It does not mean the biological repeated experiments.

A: We understand that we did not perform “biological repeated experiments” but we do not claim something like that in the MS. We grow the plants at different concentrations of carbon dioxide (which is not very easy) and we used different plants to induce the second stress (drought).

Minor editing of English language required

A: Regarding English correction, we used the Grammarly professional version for the first copyediting, and the final version of the MS was corrected by an American native English speaker (see acknowledgments). Even more, we asked him to correct the MS again.

Lucian Copolovici
